# Novel methodology for assessing total recovery time in response to unexpected perturbations while walking

Uri Rosenblum[1,2], Lotem Kribus-Shmiel[1], Gabi Zeilig[3,4], Yotam Bahat[1], Shani Kimel-Naor[1], Itshak Melzer[2], Meir Plotnik[1,5,6]*

1 Center of Advanced Technologies in Rehabilitation, Sheba Medical Center, Tel HaShomer, Israel, 2 Department of Physical Therapy, Recanati School for Community Health Professions, Faculty of Health Sciences, Ben-Gurion University of the Negev, Beer-Sheva, Israel, 3 Department of Neurological Rehabilitation, Sheba Medical Center, Tel HaShomer, Israel, 4 Department of Physical and Rehabilitation Medicine, Sackler Faculty of Medicine, Tel Aviv University, Tel Aviv, Israel, 5 Department of Physiology and Pharmacology, Sackler Faculty of Medicine, Tel Aviv University, Tel Aviv, Israel, 6 Sagol School of Neuroscience, Tel Aviv University, Tel Aviv, Israel

* meir.plontik@sheba.health.gov.il

**Data Availability Statement:** All relevant data are within the manuscript and its Supporting Information files.

## Abstract

Walking stability is achieved by adjusting the medio-lateral and anterior-posterior dimensions of the base of support (step length and step width, respectively) to contain an extrapolated center of mass. We aimed to calculate total recovery time after different types of perturbations during walking, and use it to compare young and older adults following different types of perturbations. Walking trials were performed in 12 young (age 26.92 ± 3.40 years) and 12 older (age 66.83 ± 1.60 years) adults. Perturbations were introduced at different phases of the gait cycle, on both legs and in anterior-posterior or medio-lateral directions, in random order. A novel algorithm was developed to determine total recovery time values for regaining stable step length and step width parameters following the different perturbations, and compared between the two participant groups under low and high cognitive load conditions, using principal component analysis (PCA). We analyzed 829 perturbations each for step length and step width. The algorithm successfully estimated total recovery time in 91.07% of the runs. PCA and statistical comparisons showed significant differences in step length and step width recovery times between anterior-posterior and medio-lateral perturbations, but no age-related differences. Initial analyses demonstrated the feasibility of comparisons based on total recovery time calculated using our algorithm.

## Introduction

For older adults, falls are a debilitating health problem affecting physical and psychological health [1]. Most falls in independent older adults occur when they slip (27%-32%) or trip (35%-47%) while walking [1]. Thirty percent of community-dwelling older adults fall at least once a year [1], and 10%-20% are recurrent fallers [2,3]. About 20% of falls are injurious,

**Funding:** The author MP funding for this work from the Israeli Ministry of Science and Technology - https://www.gov.il/en/Departments/Units/most_planning_and_control, grant #3-12072, and from the Israel Science Fund - https://www.isf.org.il/#/, grant #3-14527. The author UR is supported by a stipend from Ben-Gurion University of the Negev - https://in.bgu.ac.il/en/pages/default.aspx, as part of a PhD scholarship. The funders had no role in study design, data collection and analysis, decision to publish, or preparation of the manuscript.

**Competing interests:** The authors have declared that no competing interests exist.

leading to reduced mobility and independence, and increased morbidity and mortality rates [4]. Falls are associated with estimated medical costs of about $50 billion per year [5].

## Center of pressure and center of mass use for assessing postural stability

A fall is defined as a failure of the balance control system to recover from an unexpected loss of balance. Early studies addressing mechanisms underlying falls evaluated the relationships between postural sway, as measured by center of pressure (COP) displacement, and falls [6,7]. Parameters generated based on the dynamics of the COP displacement facilitate modeling mechanisms underlying balance control [8–10]. For example, recent studies quantify the entropy of COP dynamics during quiet standing [11–13] and walking [14]. These studies proposed that COP data entropy can be used to differentiate between fallers and non-fallers in standing, though it is not clear which entropy features should be used. In walking, the interaction between walking speed and duration was found to have an effect on the COP complexity.

Hof et al. [15,16] showed that during walking, balance is maintained by keeping the 'extrapolated center of mass' (xCOM; i.e., generalized center of mass (COM) variable based on the COM displacement and velocity) within the boundaries of the base of support [17]. In order not to fall as a result of a perturbation during walking, kinematic modifications in, e.g., walking speed [18], step length and width [17,19], and cadence [20], are required [21,22].

## Walking recovery processes from unexpected perturbations

In order to continue walking efficiently after a perturbation, gait kinematics have to be restored to baseline levels [18,20]. Therefore, we should expect a relatively big and quick response right after the perturbation, followed by a subtler decrease in gait variability until baseline levels are achieved. This was demonstrated for walking speed and stepping frequency [18,20].

Briefly, studies documenting the first stage of recovery from perturbation described the electromyography (EMG) patterns associated with the initial stepping response e.g., [23] and step length, step width and step time [24], as well as ground reaction force profiles [25] and xCOM in relation to the base of support [17], which regain pre-perturbation baseline values in 2–3 steps (we see the same behavior in our data, not in the scope of this paper).

While these methods for quantifying recovery time concentrated on a short time window following the perturbations, quantifying fast recovery processes (i.e., first stage of recovery), longer time widows should be the focus when studying regaining regular gait pattern processes [18,20].

Snaters et al. [20] applied rapid changes to treadmill speed and measured the dynamics of step frequency adjustments. They used a two-process model of the sum of two exponentially decaying changes to quantify the time course of fast and slow components. They showed that the first component (2–3 steps after perturbation; ~1.44 seconds) is dominant in reducing the amplitude of the response and gaining control on the COM. The second component requires longer duration (~27.56 seconds). To the best of our knowledge, no other study has proposed a systematic method for estimating the total recovery time of stepping characteristics.

## Age effect on reaction to perturbations

McIlroy and Maki [26] showed that young and older adults use different strategies to regain balance after medio-lateral perturbations in standing, and that the older adults use a greater number of steps to recover. This result was repeated in walking [27].

Older adults show slower reaction time capabilities [28] that are attributed to longer muscle activation latencies in specific muscles, including hip flexors and knee extensors prior to and during the swing phase of the reacting limb [29].

## Cognitive load and its impact on recovery from perturbations

Concurrent cognitive load was found to prolong recovery times from unexpected perturbations while walking [30–33], as well as having a negative effect on postural stability among fall-prone populations with neurological disorders and/or with prior history of falls, as compared to younger healthier groups [34]. However, task dependency was previously described. While relatively easy cognitive tasks (e.g., walking while talking) have no apparent effect on the motor task, more complex cognitive tasks (e.g., stroop task) show effects of attention [35,36]. Additionally, Ghai et al. [34] found that some dual tasks enhance (e.g., auditory switch task) and other adversely impact (e.g., complex mental arithmetic task) postural stability.

In the present study, step length and step width were calculated to accomplish the following aims: (1) devise an algorithm to calculate total recovery time; (2) conduct exploratory pilot analyses to demonstrate the feasibility of the new method in characterizing total recovery time among young and older adults under different perturbation conditions; and (3) explore differences in total recovery time for step length and step width parameters, as a function of perturbation direction, participant group (young versus older adults), and cognitive load. We hypothesized that total recovery time would be longer following medio-lateral than anterior-posterior perturbations, young adults would recover faster than older adults, and total recovery time would be shorter under single-task than dual-task conditions.

## Methods

### Participants

Twelve healthy young adults and twelve healthy older adults participated in the study (see demographic and physical characteristics in Table 1). Exclusion criteria were: (1) obesity (body mass index (kg/m2) > 30 [37]); (2) orthopedic condition affecting gait and balance (e.g., total knee replacement, total hip replacement, ankle sprain, limb fracture, etc.); (3) cognitive or psychiatric conditions that could jeopardize the participant's ability to participate in the study (Mini-Mental State Exam score < 24 [38]); (4) heart condition, such as non-stable ischemic heart disease or moderate to severe congestive heart failure; (5) severe chronic obstructive pulmonary disease; (6) neurological disease associated with balance disorders (e.g., multiple sclerosis, myelopathy, etc.).

**Table 1. Comparison of demographic and physical characteristics (mean ± standard deviation) in young and older adults.**

|  | Young Adults (n = 12) | Older Adults (n = 12) | *P* value |
|---|---|---|---|
| Age (years) | 26.92 ± 3.40 | 69.50 ± 5.20 | < 0.01* |
| Sex (F/M) | 5/7 | 6/6 | 0.56 |
| Height (cm) | 168.42 ± 7.32 | 169.67 ± 6.68 | 0.82 |
| Weight (kg) | 63.67 ± 10.26 | 78.34 ± 16.22 | 0.02* |
| BMI (kg/m$^2$) | 22.35 ± 2.47 | 27.13 ± 4.82 | <0.01* |
| Years of education | 15.25 ± 2.67 | 14.27 ± 2.24 | 0.66 |

Comparisons were made using the Mann–Whitney U test and cross-tabulation as needed; BMI, body mass index;

*$p < .05$.

All participants signed an informed consent prior to entering the study, which was approved by the Sheba Medical Center Institutional Review Board (Approval Number 9407–12).

## Apparatus

We implemented a VR-based paradigm using the Computer Assisted Rehabilitation Environment (CAREN) High-End (Motek-Medical, the Netherlands; see Fig 1), a motion base platform with an imbedded treadmill, surrounded by a 360˚ dome-shaped screen, enabling optimal immersion in a largescale VR setting. During walking trials, participants (in a safety harness) were presented with simulated ecological surroundings (e.g., urban public park). Underneath the belts there are two parallel force plates (Zemic load cells; The Netherlands), one under each belt. Each force plate has six sensors. Two sensors for the X axis (medio-lateral; ML), three sensors for the Y axis (vertical), and one sensor for the Z axis (anterior-posterior; AP). Force measurement precision is 0.5N (~51gr) and the reliable range of measurement is 0-500kg.

**Self-paced walking mode.** The treadmill was operated in self-paced mode, in which the motor is operated as a function of the instantaneous position and speed of the participant, which are sampled using a motion capture system (see more details in [39]).

**Physical perturbations.** Perturbation types were classified according to the following criteria: (1) perturbation direction–medio-lateral (i.e., right or left platform displacement) or anterior-posterior (i.e., deceleration of one of the treadmill belts); (2) gait phase–immediately after initial contact, mid-stance, or towards toe off, as detected in real time from foot markers and force plates data; and (3) foot of reference (right or left foot that were detected in the relevant gait phase). An example for perturbation type by criteria would be: (1) platform left in (2) initial contact (3) right = a displacement of the platform to the left was executed when initial contact phase was detected for the right foot. Medio-lateral platform perturbations were achieved by displacing the moving platform 15 cm, to the left or to the right, over 0.92 seconds (perturbation Level 12, see Fig 2). The platform held its new position for 30 seconds, to allow full recovery of the gait parameters (i.e., step length and step width), and then returned gradually to its original position over three seconds; Anterior-posterior treadmill belt perturbations were achieved by reducing the speed of one of the treadmill belts by 1.2 m/s with a deceleration

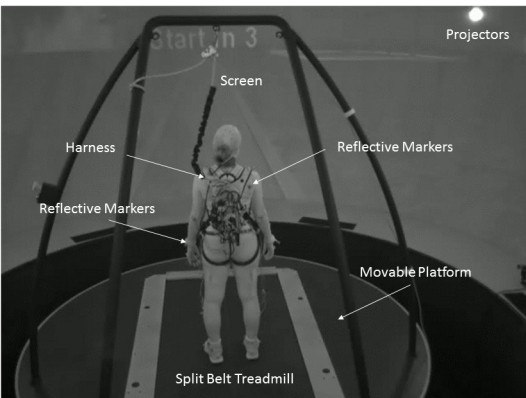

**Fig 1. Experimental setup.** Participants walked on a treadmill in self-paced mode. The treadmill is embedded in a moveable platform with six degrees of freedom (three translations and three rotations). They were exposed to unexpected platform (medio-lateral) or treadmill (anterior-posterior) perturbations in a virtual reality environment. Vicon motion capture cameras were used to calculate walking parameters of step length and step width based on marker data from the feet. A small backpack was used to carry a wireless amplifier connected to biosensors (e.g., electroencephalography, electromyography), data from which are not reported here.

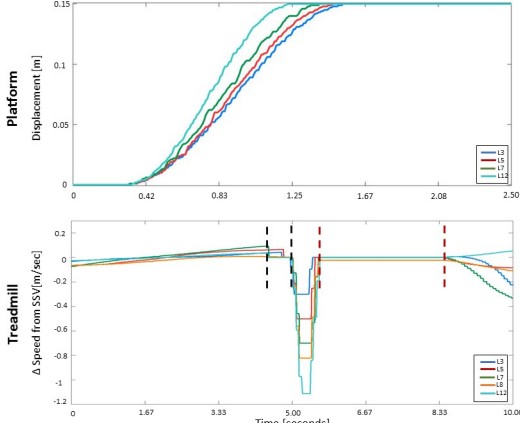

**Fig 2. Perturbation profiles used in the platform (top) and treadmill (bottom) experiments.**

of 5 m/s$^2$ (perturbation Level 12, see Fig 2). To enable anterior-posterior perturbations, the treadmill speed was fixed to the measured average self-paced walking speed five seconds prior to perturbation onset. The perturbation was presented when the relevant gait phase was detected (see below), and three seconds later the treadmill was once again set to self-paced mode (see perturbation profiles, Fig 2).

We aimed to introduce perturbations in particular phases of the gait cycle: initial contact, mid-stance, and toe off. Relevant gait phase was detected online using force plate and toe marker data in the sagittal plane. The former was filtered using a low-pass filter with cutoff frequency of 5Hz and a force threshold of 25N was set. Initial contact was detected when the vertical force from the foot in front (leading limb; identified using the toe marker) exceeded a minimum of 35N (to avoid force plate baseline noise); mid-stance was detected 200 ms after initial contact; and toe off was detected when the vertical force from the force plate on the side of the trailing limb was below 5 N. Once the relevant gait phase was detected, a perturbation execution command was initiated by the system. Since there was a delay between the detection of the relevant gait phase and perturbation execution (~250 ms), we re-evaluated the actual perturbation type (i.e., initial contact, mid-stance, or toe off) post-hoc. Gait phases and leading limb were defined using the heel markers' position in the sagittal plane [40]. Initial contact and toe off were detected when the heel marker was at its maximal forward or backward position, respectively, with 0.025 second margins. Perturbation that was introduced beyond these margins was considered a mid-stance perturbation.

While the intention was to introduce perturbations at magnetite Level 12 (see Fig 2), four older adults expressed fear when perturbation magnitude was raised, and were tested using lower magnitudes (levels 3–8, see Fig 2 legend). Perturbation magnitude and timing (see experimental procedure below) were controlled and modified by the VR system computer.

Possible perturbation levels ranged from Level 1 to Level 20, participants in the study were subjected to perturbation magnitude Level 12. For platform perturbations, Level 12 is implemented by displacing the platform 15 cm to the right or left over 0.92 seconds. The levels were used to allow for participants who feared the high magnitude perturbations to participate (four of the 12 older adults). The levels were constructed so that the platform displacement is always 15 cm, and the time to complete the distance was manipulated between 1.36 seconds (Level 1) and 0.6 seconds (Level 20); decrease of 0.04 seconds per level. For treadmill perturbations, Level 12, we used a deceleration of 5 m/s$^2$ and speed reduction of 1.2 m/s. For the different levels we used fixed deceleration of 5 m/s$^2$, and the reduction in speed was manipulated between

0.1 m/s (Level 1) and 2 m/s (Level 20); increase of 0.1 m/s per level. Prior to and immediately following the treadmill perturbation (periods between black and red vertical dashed lines, respectively), self-paced was switched off and the two belt speeds were fixed to the mean walking speed in the last five seconds; (see text for further details). SSV–steady state velocity

**Capturing gait parameters.** Gait speed was obtained from an encoder on the drive shaft of the treadmill motor. Spatial-temporal gait parameters (i.e., step times, step length, and step width) were obtained from a Vicon motion capture system (Vicon Motion Systems, Oxford, UK) (set of 41 body markers, 18 cameras, 120Hz sampling rate, 0.11 cm spatial accuracy). Gait phases were detected using foot marker and force plate data.

## Experimental procedure

**Stage I: Self-paced walking–learning and acclimation.** The trials started with acclimation sessions during which participants were introduced to the self-paced walking paradigm. Participants were asked to maintain their comfortable walking speed for one minute, and then to dynamically modify their gait speed, including accelerating, decelerating, coming to a full stop, and resuming their comfortable walking speed. This was repeated until they felt confident and performed the transition from standing to comfortable walking speed smoothly.

**Stage II: Introducing physical perturbations during walking.** The protocol consisted of walking trials in two different scenes (park and a no-hands hanging bridge over a deep canyon). Each scene was used in four walking trials of single-task and dual-task conditions (see Fig 3 for more details).

Once the participant reached steady state walking velocity (see S1 Appendix), one of several types of unexpected perturbations was randomly presented (see below) in two trial conditions: (a) the single-task condition, in which no additional cognitive load was introduced; (b) the dual-task condition, in which participants were asked to perform concurrent arithmetic calculations [41], higher cognitive load. Similar arithmetic tasks were practiced by the participants prior to the gait trials, while sitting and standing. Participants were instructed to react naturally to the perturbations, to prevent themselves from falling.

A total of 18 possible surface perturbations were used in random order to reduce learning effects anticipatory reactions from trial to trial [21,42].

At least 30 seconds of baseline walking were implemented before and after every perturbation to allow full recovery of the kinematic gait parameters. This experiment was part of a

## Study protocol

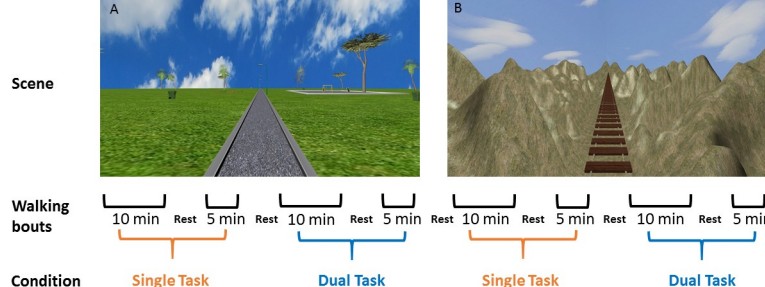

**Fig 3. Study protocol.** The protocol consisted of walking trials in two different scenes, park (A) and a no-hands hanging bridge (B). Each scene was used in four walking trials of single-task and dual-task conditions in walking bouts of ten and five minutes with rest between walking trails. Participants performed eight walking trails, with the exception of one young and five older participants who asked to stop. The order of the scenes and the conditions were randomized across participants.

larger study with the overall aim of developing algorithms for the analysis of physiological networks for fall prevention in elderly subjects with and without neurological disease using virtual reality environments. Additional procedures related to the larger experimental protocol are described in the S1 Appendix.

## Data handling and outcome calculations

The proposed algorithm for the estimation of total recovery time for gait (see below) was based on the re-stabilization of discrete step length and step width variables.

**Step width and step length calculation.** A MATLAB Graphical User Interface (MathWorks Inc.) was customized to assess gait cycle parameters. Motion capture data from heel markers were used to semi-automatically detect initial contact times and calculate step length and step width (see Fig 4A for more details). The following equations (conducted for each leg) were used for the calculation of step length and step width, respectively:

$$SL = HM_L(y) - HM_T(y) \tag{Eq 1}$$

$$SW = HM_L(x) - HM_T(x) \tag{Eq 2}$$

where SL is step length, SW is step width, y is anterior-posterior axis, x is the medio-lateral axis, $HM_L$ is the leading limb heel marker at the moment of initial contact, and $HM_T$ is the trailing limb heel marker at the moment of initial contact of the leading limb. Examples of actual step length and step width data before and after perturbations are shown in Fig 4B and 4C, respectively.

**Definition of an automated algorithm to detect total recovery time.** Given a graph that represents a vector of gait parameter data (i.e., step length or step width) with a perturbation at time t, the goal of the algorithm is to detect the first point after t at which the graph (i.e., the

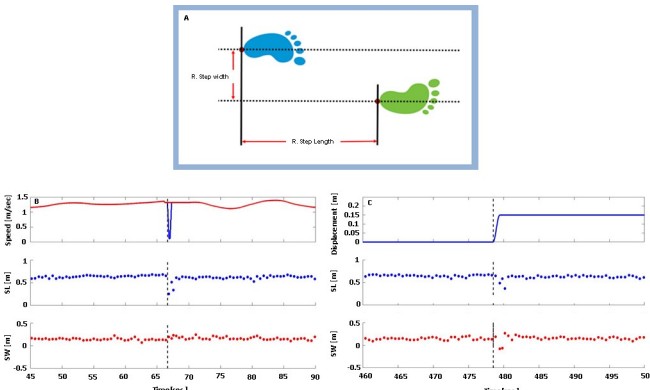

**Fig 4. Step width and step length calculation and sample data.** (A) Illustration of step length and step width calculation. (B) Step length and step width behavior in response to anterior-posterior treadmill belt perturbation. Top panel: trace of treadmill belt speed shows treadmill perturbation profile; red–right belt, blue–left belt (seen only at time of perturbation, as the two belt speeds were otherwise identical); middle panel: trace of step length values prior to and post perturbation; lower panel: trace of step width values prior to and post perturbation. (C) Step length and step width behavior in response to medio-lateral platform perturbation. Top panel: trace of platform displacement shows platform perturbation profile (positive values = right direction); middle panel: trace of step length values prior to and post perturbation; lower panel: trace of step width values prior to and post perturbation. Negative step length values represent backward steps of the recovering limb, while negative step width values represent a crossover step with the leading limb. (B) and (C) depict 45 and 40 seconds, respectively, of walk trials from one participant. Values from the right and left leg are depicted in all panels (alternating; not distinguished in the trace). Vertical dashed line represents perturbation onset time.

values in the vector) is considered "recovered". Meaning having reached a stable pattern–the variability of the gait parameter is reduced to baseline levels.

Fig 5 will be used to facilitate our explanation of the algorithm, which operates in three stages: (1) computes two implied graphs using a moving window of six samples, a graph of means, and a graph of standard deviations, from the original data; (2) standardizes the implied graphs, and creates a combined graph; and (3) scans the combined graph from left to right and continually updates an index, which eventually represents the point of recovery (see Fig 5 and S1 Appendix).

**Black dashed line represents perturbation onset.** *Stage 1*. The original walking parameter graph (Fig 5A) is scanned using a moving window of six samples to compute two graphs: mean and standard deviation (Fig 5B). Each point in the graphs represents the mean or standard deviation of the six original samples prior to it.

*Stage 2*. For each implied graph, 20 samples prior to perturbation onset were taken as baseline ($\mathbf{M_{BL}}$ and $\mathbf{SD_{BL}}$; Fig 5) to standardize the implied graphs. Mean and standard deviation were calculated for each baseline. Then, each value $M_i$ and $SD_i$ of the implied graphs is standardized, and matching standardized data points are summed to obtain the overall sum of deviations (OSDev–Fig 5C) using the following equation:

$$\text{OSDev}, i = 0.25 * \left| \frac{Mi - mean(\mathbf{M_{BL}})}{stdev(\mathbf{M_{BL}})} \right| + max\left(0, \frac{SDi - mean(\mathbf{SD_{BL}})}{stdev(\mathbf{SD_{BL}})}\right) \qquad \text{(Eq 3)}$$

where i = 1, 2, . . ., L; and L = length of implied graphs. Intuitively, the combined graph estimates the deviation from the behavior of the graph prior to perturbation. The second element in the equation takes the positive values and replaces the negative values with zeros, as negative values mean lower standard deviation, which corresponds to smaller variance (i.e., more stable). Since the average value is less indicative of gait recovery, the elements in Eq 3 are summed

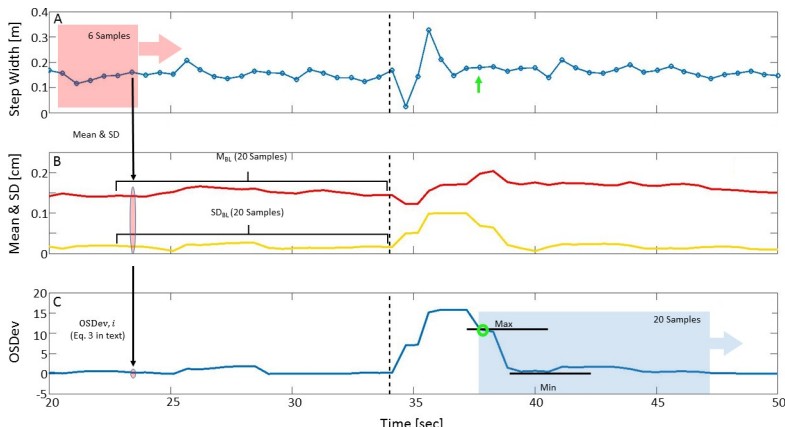

**Fig 5. Algorithm performance steps.** (A) Graph of step width values (blue circles; connected to graphically enhance perturbation effect). (B) Implied graphs for means (M; red) and standard deviations (SD; yellow), calculated using a moving window of six samples. Each point in the graphs represents data from six preceding original data points (depicted by the black vertical arrow between panel A and B). For each implied graph, 20 samples prior to perturbation onset were taken as baseline ($\mathbf{M_{BL}}$ and $\mathbf{SD_{BL}}$, denoted by horizontal black brackets). (C) $\mathbf{M_{BL}}$ and $\mathbf{SD_{BL}}$ were used to standardize the implied graphs, to create the overall sum of deviations (OSDev–blue line, calculated using Eq 3). To determine the recovery point, the algorithm scanned the OSDev graph from left to right, using a moving window of 20 values starting at perturbation onset. The recovery point is then defined using a mathematical criterion (see S1 Appendix) that assesses minimal changes in signal amplitudes (green circle; corresponds to the original step value indicated by the green arrow in panel A).

up with a ratio of 1:4 in favor of the second element. The outcome of this stage is the OSDev graph (Fig 5C), which is used to inspect the recovery criteria.

*Stage 3*. To determine the point of recovery, the algorithm scans the OSDev graph from left to right, using a moving window of 20 values starting at perturbation onset. For each window, the algorithm computes "amplitude", defined as the maximum difference between any two values in the window ($cur_{amp}$). Two more amplitudes are considered: (1) $first_{amp}$ = the amplitude of the first window right after perturbation (the a priori assumption is that this is the largest amplitude difference due to the reaction to perturbation); and (2) $best_{amp}$ = an amplitude that is at least 50% smaller than the first amplitude and satisfies the condition in Eq 4. The condition for updating $best_{amp}$ is reevaluated in every window using the $cur_{amp}$ value. $cur_{ind}$ and $best_{ind}$ are the indeces of the first sample in the window for $cur_{amp}$ and $best_{amp}$, respectively, and are updated with them as needed. The point of recovery is the point at which the amplitude difference is small enough relative to the "distance" from perturbation (see Eq 4) and is found in the $best_{ind}$ after the last window was assessed.

$$\frac{best_{amp} - cur_{amp}}{first_{amp} - best_{amp}} > 1\% \cdot (cur_{ind} - best_{ind}) \qquad \text{(Eq 4)}$$

The condition in Eq 4 is stricter if the position of the current window is further away from the $best_{amp}$ window index. This is done in order to overcome the natural variance of amplitudes. The point of recovery is defined as the $best_{ind}$ after calculating all amplitude differences in the whole OSDev vector within consecutive windows (see Section 4 S1 Fig in S1 Appendix for complete description of the process of defining the point of recovery).

It should be clarified that the algorithm looks for the point in time where the window's amplitude stops decreasing. This is done by considering the significance of the difference from the previously detected amplitude decrease. For this we used Eq 4, and we set a conservative threshold of 1% to prevent undesired delays in total recovery time detection.

## Data handling and statistical analyses

Non-parametric statistics were used. Medians and 95% confidence interval of the median were used to describe central tendency of the total recovery time distributions.

ICC estimates and their 95% confident intervals were calculated for total recovery time detection using SPSS statistical package version 23 (SPSS Inc, Chicago, IL) based on a mean-rating (k = 2), absolute-agreement, 2-way mixed-effects model, calculated from total recovery time estimation of 100 random perturbations (200 samples– 100 for step length and 100 for step width), comparing the algorithm for total recovery time detections with human labeling (UR, blinded to the results of the algorithm).

To explore the effects of perturbation direction (medio-lateral versus anterior-posterior), the group (young versus older adults), and cognitive load (single task versus dual task) on total recovery times, principal component analysis (PCA) was used, applying the singular value decomposition (SVD). PCA allows for clustering of multidimensional data, in our case, comparing total recovery time, a bi-dimensional property consisting of the combination of step length and step width, across different groups, conditions, and perturbation direction. Kruskal-Wallis one-way ANOVA was used to test the effect of perturbation intensity on total recovery time. Since we found a significant effect of perturbation intensity on step length total recovery time (p<0.01), only Level 12 perturbations (87% of all perturbations) were included in the PCA analysis for comparisons of total recovery time for group, condition, and perturbation direction. Prior to using PCA, values of total recovery time for step length and step width were standardized (i.e., transformed into units of standard deviation), using Eq 5, so they

could be comparable.

$$(tRcT - \overline{tRcT})/std(tRcT) \qquad\qquad (Eq\ 5)$$

were $tRcT$ is the total recovery time in seconds for step length or step width, and $std$ is the standard deviation. The standardized total recovery time is unitless. These values were used in the PCA algorithm as an n x 2 matrix (were n is the length of the standardized step length and step width vectors).

PCA was performed using MATLAB R2016b (MathWorks Inc.). Wilcoxon rank sum test was used to compare central tendencies along the different principal components (PC) to test the effects above. Additionally, Wilcoxon rank sum test was used to test for differences between groups, conditions, and perturbation direction for total recovery times of step length and step width separately. Significance level was set to p<0.05.

The protocol is also published here: dx.doi.org/10.17504/protocols.io.bdjvi4n6.

## Results

### Algorithm performance and reliability

Total recovery time values were calculated for step length and step width separately, based on 829 samples. "No deviation" was found in 47 and 57 perturbations for step length and step width, respectively (11 overlapping). "No recovery" was detected in 11 and 33 perturbations for step length and step width, respectively (1 overlapping; see Fig 6).

The 829 perturbations include samples from pilot trials aimed at defining the appropriate baseline period (>18 seconds of undisturbed walking prior to perturbation to include at least 26 steps–see S1 Appendix).

**Total recovery time distributions.**   Distribution of total recovery time in the two groups, for step length and step width, across all participants, experimental conditions, perturbation types and intensity Level 12 is presented in Fig 7A–7D. In most cases, participants recovered from the unexpected mechanical perturbations after 4–6 seconds regardless of the perturbation type (Fig 7A–7D). The 95% confidence intervals of the medians of total recovery time were 4.70–5.06 seconds for step length in older adults, 5.32–5.96 seconds for step width in older adults, 4.86–5.14 seconds for step length in young adults and 5.49–5.97 seconds for step width in young adults (S1 and S2 Tables summarizes all results for Level 12 perturbation types).

The detection algorithm results in comparison to human labeling showed moderate reliability (ICC = 0.57, 95% CI = 0.42–0.67, p<0.01).

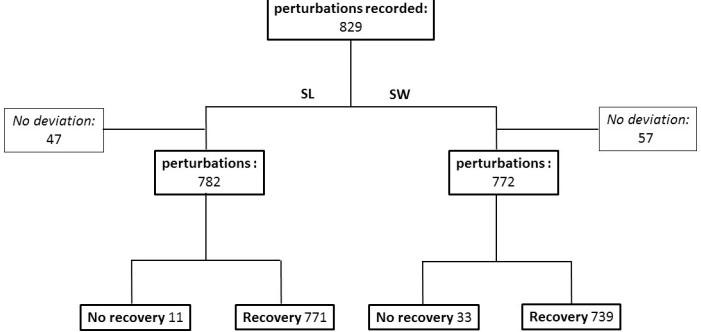

**Fig 6. Flow chart of algorithm performance for gait parameters of step length (SL) and step width (SW).**

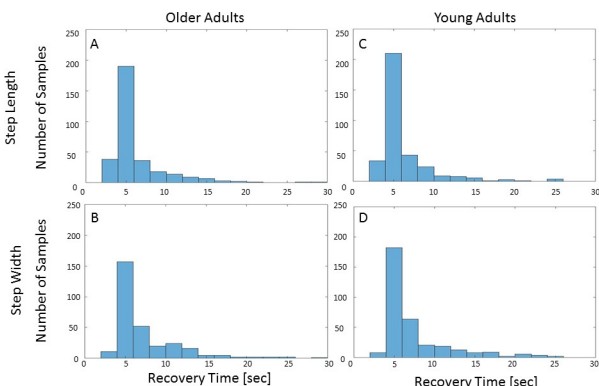

**Fig 7. Total recovery time distributions. Bins of two seconds were used.** (A), (C), distribution of step length total recovery time for older and young adults, respectively. (B), (D), distribution of step width total recovery time for older and young adults, respectively. The histograms show a non-normal distribution with a skew to the right of total recovery time for young and older adults.

**Exploratory analyses: Effects of perturbation type, group, and cognitive load.** PCA plots comparing total recovery times for step length and step width are shown in Fig 8 and in Table 2. Statistically significant differences were found between total recovery time values in response to medio-lateral versus anterior-posterior perturbations–for step length median total recovery time was 5.14 and 4.61 seconds from ML and AP perturbations, respectively. For step width, the corresponding results were 5.52 and 6.41 seconds, respectively. No significant differences were found between older and young adults or between cognitive task conditions. Furthermore, the plots show similar variability in each pair of calculated principal components (e.g., 53% and 47% for PC1 and PC2, respectively, Fig 8B), which means that each PC of the transformed total recovery time values for regaining regular, stable step length and step width parameters contributed roughly equally to the ability to find differences in the above comparisons.

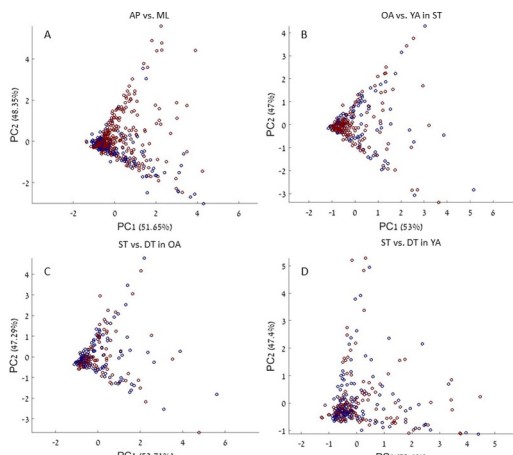

**Fig 8. Principal component plots of total recovery time values.** Principal component plots of total recovery time values for regaining stable gait parameters (step length and step width) for the following comparisons: (A) perturbation direction–anterior-posterior (blue dots) versus medio-lateral (red dots); (B) group–older adults (blue dots) versus young adults (red dots); (C) cognitive load conditions–single task (blue dots) versus dual task (red dots) among older adults; and (D) cognitive load conditions–single task (blue dots) versus dual task (red dots) among younger adults. PC1 and PC2 are the two components found by the PCA, and the percentages represent the percent of variation explained in the data by each component (see scree plots, S2 Fig, in S1 Appendix).

**Table 2. Mean standardized total recovery time for step length and step width to standard deviation units (standardized mean): Independent samples comparisons of perturbation direction, participant group, and cognitive load for the first and second principal components.**

| | | ML vs. AP | | OA vs. YA | | ST vs. DT in OA | | ST vs. DT in YA | |
|---|---|---|---|---|---|---|---|---|---|
| | | AP | ML | OA | YA | ST | DT | ST | DT |
| PC1 | Standardized mean | 0.11 | -0.05 | 0.03 | -0.04 | 0.02 | -0.10 | -0.05 | 0.08 |
| | Z Value | 0.24 | | 0.15 | | 0.62 | | -1.23 | |
| | P value | 0.81 | | 0.88 | | 0.54 | | 0.22 | |
| PC2 | Standardized mean | -0.39 | 0.13 | 0.01 | -0.03 | 0.02 | -0.02 | 0.00 | 0.02 |
| | Z Value | -6.21 | | 0.81 | | 0.66 | | -0.19 | |
| | P value | <0.01* | | 0.42 | | 0.51 | | 0.85 | |

SL = step length; SW = step width; PC = principal component; AP = anterior-posterior; ML = medio-lateral; OA = older adults; YA = young adults; ST = single task; DT = dual task.

*$p < .05$.

SL = step length; SW = step width; OA = older adults; YA = young adults; AP = anterior-posterior; ML = medio-lateral; ST = single task; DT = dual task; PC = principal component.

In secondary analysis, Wilcoxon rank sum test was used to test for differences between groups, conditions, and perturbation direction for total recovery times of step length and step width separately. No significant differences were found for any of the comparisons (p>0.1).

## Discussion

Our newly developed algorithm successfully estimated total recovery time in 91.07% of the samples, detected no deviation in 6.27%, and could not converge on the available data points in 2.65% (Fig 6). Across groups, conditions, and perturbation types, step length and step width regained stability within the first six seconds after perturbation (Fig 7). An exploratory PCA identified a significant difference between anterior-posterior and medio-lateral perturbations but no significant difference between older and young adults or between single-task and dual-task conditions (Table 2). In comparison to human labeling, the algorithm shows moderate ICC values (see more in the Limitations section below).

### Total recovery time from physical perturbations—earlier estimations

Response times to physical perturbations during walking were previously described based on muscle recruitment (EMG findings) (e.g., [42,43]) and on first recovery step (e.g., "crossover" or "lateral" step) (e.g., [17]). These studies indicated relatively fast responses (~100–350 ms and >300 ms, respectively). In the present study, we assessed the time it took to regain stable gait parameter patterns after surface perturbations, and found that it was 10–20 times longer than initial step responses.

Hof et al. [17] reported that two steps should be sufficient to contain the extrapolated center of mass in the boundaries of the base of support after a perturbation while walking. Lee et al. [25] found similar results when analyzing COP data, trunk kinematics, and center of mass velocity to characterize the recovery responses from a trip in balance-impaired individuals who are prone to falling. We believe this is only a partial view of the recovery process, as some gait parameters, such as speed, cadence [18,20], step length, and step width (in our study) take longer to recover. In the case of a perturbation, such as a "slip" (i.e., loss of footing of the stance foot) or a "trip" (i.e., obstruction of the swinging foot), the extrapolated center of mass might cross the base of support, implying instability. Thus, a recovery strategy such as trunk

movement or a step in the same direction and of equal magnitude to the displacement of the extrapolated center of mass is essential to keeping equilibrium [17,19,25]. This mechanism leads to rapid containment of the extrapolated center of mass within the base of support, and prevents a fall. We believe this first recovery process is followed by a secondary process of gait optimization, where gait parameters are fine-tuned. This is supported by the findings of Snaters et al. [20] who studied the recovery of step frequency after a perturbation. They observed two stages in stepping frequency recovery process: (1) fast and dominant change, occurring in ~1.44 seconds (2–3 steps); (2) the second stage takes about 27 seconds to complete. We attribute the discrepancy between our total recovery time results (5–6 seconds) and the time Snaters et al. report for the second stage mainly to methodological factors: (1) they evaluate cadence, while we evaluate step width and step length, imperative parameters for balance recovery and the containment of COM in the boundaries of the base of support; (2) type of perturbations: we used abrupt, physical changes (slips, displacements), while they used a smooth, longer transition to a new constant speed, which leads to a longer gait adaptation process.

Our results differ from those reported by O'Connor and Donelan [18], who showed that people gradually return to steady state baseline walking speed in ~365 seconds after a visual walking speed perturbation. We believe that the discrepancy between these findings and our own is directly related to the paradigm, as visual stimuli (perturbations) are known to elicit a delayed response time (5.7 seconds, in this case) [44]. Another factor differentiating the studies is duration of perturbation. Our participants were exposed to a relatively short, discrete perturbation, while O'Connor and Donelan introduced either one gradual perturbation or a series of sinusoidal perturbations. We believe that in the latter case, continuous processes of adaptation to the inherently unstable environment delay the stabilization process. Finally, changes in step length affect walking speed, which can explain the shorter total recovery times in the present study (4–6 seconds in the majority of cases, see Fig 7).

## Total recovery time distribution and variability

Figs 7 and 8 suggest high variability in total recovery times, with a central tendency skewed to the right. Recovery stepping responses to unexpected perturbations are constrained by a floor effect, as there is a limit to the minimal time required to react. In addition, participants had to respond promptly in order to prevent a fall (first recovery process), pushing the median towards lower values. However, after the initial recovery step response, gait amendments are not essential to prevent a fall, and the process of regaining a stable pattern of gait parameter generation lasts longer. The price paid for slow gait recovery is marginal (i.e., more energy expenditure [45,46]) and can therefore be overlooked at times.

## Age, task, and gait parameter effects on total recovery time–pilot feasibility analyses

Non-parametric statistics, PCA, were used to demonstrate the feasibility of using the total recovery time parameter, which was obtained from the new algorithm presented in the present work, for studying the differences across different cohorts, conditions, perturbation directions, and gait parameters. We view these results as preliminary and limited by small sample size. Herein these results are discussed in comparison to the existing literature.

**Young versus older adults.** We found no significant differences in the total recovery times between young and older adults. The non-significant difference we found is in agreement with McIlroy and Maki [26] that characterized the compensatory stepping responses of older and young adults to unpredicted postural perturbations. They show that stepping limb kinematics are very similar between the groups, and that young and older adults react in

similar times. Byrne et al. [47] found that young and older adults differ in gait coordination only in the breaking period, no differences were found in constant walking. Liu and Lockhart [48] showed that older adults use alternative strategies to compensate for physiological deficits to maintain balance, achieving the effectivity of young adults. Similar findings were presented for neural resource allocation [49]. Additionally, we only compared older adults who could withstand the perturbation magnitude Level 12, the same as the young adults. By doing so, we only used data from the "better" older adults. It might be the case that if we would compare the "weaker" older adults to the young adults, a group effect will be significant.

**Single task versus dual task.** Our preliminary analyses revealed no differences in the total recovery times, whether a concurrent cognitive task was introduced or not. Previous studies showed that young and older adults are affected by concurrent dual task, but only if it is challenging to them [35,36]. Further study is required to delineate the effects of cognitive load on total recovery time, for example, by assessing how confounding performance levels of the congruent task are on the total recovery time results.

**Medio-lateral versus anterior-posterior.** We found significant differences between total recovery time from ML perturbation as compared to AP perturbations: values for step length recovery time were higher after ML, and recovery time of step width were higher for AP perturbations.

One may argue that this finding is counter intuitive, since AP perturbations have a greater physical impact on step length, and this parameter should recover more slowly, and ML perturbations immediately challenge ML control of COM placement within the base of support, therefore step width should recover more slowly.

We speculate that since regulation of gait parameters is prioritized in the direction of the more disturbed gait parameter (e.g., step width in the case of ML perturbation), the regulation of other parameters is delayed. This speculation is in line with findings about general motor control principles. For example, Todorov and Jordan [50] formulated the "minimal intervention principle" with reference to the performance of manual reaching movements. According to this principle, when a motor task involves controlling multiple degrees of freedom at once, the system optimizes motor coordination by reducing the degrees of freedom being attended to at any given moment, and only relevant variability is reduced, to allow for optimal task performance. In the case of response to perturbation, the variability in the more perturbed plane (i.e., step width in the ML perturbations) is thus regulated first.

Finally, we re-iterate that our hypotheses should be corroborated by additional data to be collected in future studies.

## Algorithm advantages

Our method calculates total recovery time from unexpected physical perturbations during human walking. It can be used to find the point of recovery with respect to any discrete variable for which sufficient data points prior to and post perturbation are available. Other algorithms in the literature are specific to one gait parameter (e.g., gait speed [18] or step frequency [20]), which are continuous in their nature, not series of discrete events. It is relatively simple, based on an algorithm that uses mean, standard deviation, and amplitude differences as independent variables. Finally, the major advantage of this algorithm is that it has the ability to adjust, per case, considering the baseline before perturbation.

## Algorithm engineering advantages

The algorithm can be easily manipulated to fit different types of data. As it is now, the algorithm requires at least 26 data points for baseline, and at least 20 data points after the

perturbation for just the first window. If these conditions are not met (e.g., due to the nature of the data), the window size of the baseline for the normalization of mean and standard deviation can be shortened. The window for finding the point of recovery on the OSDev graph can also be shortened. The right side of Eq 4 defines the algorithm's sensitivity to variation in the data. When the variability is high and point of recovery designation is too late, the constant on the right side of Eq 4 can be increased. The percent change from the first$_{amp}$ for the initiation of best$_{amp}$ and best$_{ind}$ can be decreased as well.

### Implications, limitations, and future directions

In the present study, we calculated recovery of step length and step width to a steady state. Our results suggest that beyond the quick balance recovery after physical perturbations (<2 seconds [17,20,24,25]), there are additional recovery processes regulating gait patterns. This regulation might be occupying some brain capacity and elevating energy expenditure.

Since there is no "gold standard" for detecting total recovery time, we compared the algorithm performance with human labeling; this comparison is limited since human labeling is subjective.

As discussed, direct follow up study is required to assess age, task, and step length/width effects. Further, the assessment of total recovery time may facilitate understanding of the mechanisms related to gait regulations, e.g., brain activity as expressed by electroencephalography and energy expenditure.

Future studies are needed to integrate a greater number of gait parameters to further understand adaptation process related to the total recovery time.

We recommend at least 30 seconds of stable baseline prior to exposure to unexpected perturbations in future studies. Similarly, permitting more than 30 seconds after perturbation (prior to preparation for a new perturbation) might increase the likelihood of detecting total recovery time in cases where "no recovery" would otherwise be designated. Our study can pave the way to incorporating total recovery time after unexpected loss of balance as a tool in the assessment of walking and balance impairments, to aid in diagnosis and in monitoring of treatment outcomes.

### Supporting information

**S1 Appendix. Supplementary materials for methods and results.**
(DOCX)

**S1 Table. Summary of total recovery time for mid-stance, Level 12, perturbations.**
(DOCX)

**S2 Table. Summary of total recovery time for initial contact and toe off, Level 12, perturbations.**
(DOCX)

**S1 Data. Recovery times for step length and step width.**
(XLSX)

**S2 Data. Data for PCA.**
(XLSX)

### Author Contributions

**Data curation:** Uri Rosenblum.

**Formal analysis:** Uri Rosenblum.

**Funding acquisition:** Meir Plotnik.

**Investigation:** Uri Rosenblum.

**Methodology:** Uri Rosenblum, Itshak Melzer, Meir Plotnik.

**Project administration:** Uri Rosenblum, Meir Plotnik.

**Resources:** Uri Rosenblum, Yotam Bahat.

**Software:** Uri Rosenblum, Lotem Kribus-Shmiel, Shani Kimel-Naor.

**Supervision:** Itshak Melzer, Meir Plotnik.

**Validation:** Uri Rosenblum.

**Visualization:** Uri Rosenblum, Shani Kimel-Naor.

**Writing – original draft:** Uri Rosenblum, Lotem Kribus-Shmiel, Itshak Melzer, Meir Plotnik.

**Writing – review & editing:** Gabi Zeilig, Yotam Bahat, Shani Kimel-Naor, Itshak Melzer, Meir Plotnik.

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
