## [Decision Letter · Decision Letter 0]

31 Jan 2020

PONE-D-19-31620

Novel methodology for assessing total recovery time in response to unexpected perturbations while walking

PLOS ONE

Dear Dr. Plotnik,

Thank you for submitting your manuscript to PLOS ONE. After careful consideration, we feel that it has merit but does not fully meet PLOS ONE’s publication criteria as it currently stands. Therefore, we invite you to submit a revised version of the manuscript that addresses the points raised during the review process.

We would appreciate receiving your revised manuscript by Mar 16 2020 11:59PM. To enhance the reproducibility of your results, we recommend that if applicable you deposit your laboratory protocols in protocols.io, where a protocol can be assigned its own identifier (DOI) such that it can be cited independently in the future. For instructions see: http://journals.plos.org/plosone/s/submission-guidelines#loc-laboratory-protocols

We look forward to receiving your revised manuscript.

Kind regards,

Yih-Kuen Jan, PhD, University of Illinois at Urbana-Champaign

Journal Requirements:

Reviewers' comments:

Reviewer's Responses to Questions

**Comments to the Author**

1. Is the manuscript technically sound, and do the data support the conclusions?

Reviewer #1: Yes

Reviewer #2: Partly

Reviewer #3: Yes

2. Has the statistical analysis been performed appropriately and rigorously? 

Reviewer #1: No

Reviewer #2: No

Reviewer #3: Yes

3. Have the authors made all data underlying the findings in their manuscript fully available?

Reviewer #1: Yes

Reviewer #2: No

Reviewer #3: Yes

4. Is the manuscript presented in an intelligible fashion and written in standard English?

Reviewer #1: Yes

Reviewer #2: Yes

Reviewer #3: Yes

5. Review Comments to the Author

Reviewer #1: The authors of this manuscript developed a new algorithm to estimate the response time after various types of perturbations and compared the differences between young and old adults with principal component analysis (PCA). However, there are still some questions that need to be clarified:

INTRODUCTION

Overall, the introduction section introduces some previous methods regarding assessing response time. There are a lot of methods to evaluate postural stability, some of which have been described in the current article. I suggest authors cited some references about using center of mass (COM) and center of pressure (COP) to assess postural stability.

Recommended references:

• Fino, Peter C., Ahmad R. Mojdehi, Khaled Adjerid, Mohammad Habibi, Thurmon E. Lockhart, and Shane D. Ross. "Comparing postural stability entropy analyses to differentiate fallers and non-fallers." Annals of biomedical engineering 44, no. 5 (2016): 1636-1645.

• Liau, Ben-Yi, Fu-Lien Wu, Chi-Wen Lung, Xueyan Zhang, Xiaoling Wang, and Yih-Kuen Jan. "Complexity-Based Measures of Postural Sway during Walking at Different Speeds and Durations Using Multiscale Entropy." Entropy 21, no. 11 (2019): 1128.

Moreover, I suggest the authors mention some limitations about the traditional methods that estimate recovery time in response to unpredictable perturbations. It would help you justify the purpose of this study.

Another concern to me is what the reason the authors compared the differences between the young and old population with PCA. According to the research questions raised in this study, traditional ANOVA or similar statistical methods could be used for this situation.

Line 55: “…a two-stage process [[11] [12]. The…” Please delete the extra left bracket.

Line 59: “first recovery step) [e.g., [13]]. However,…” Please delete “e.g.”.

METHODS

Line 97: Please add unit for body mass index.

Table 1: Please make the decimal places consistent.

Figure 2: Please write the full name of SSV.

Line 160: Please specify how many cameras the study used.

Line 188: What does the “foot of reference” mean?

Line 234-238: I recommend stating the number of samples of a moving window in text.

Line 281-290: Please specify the p value used in this study.

RESULTS

For PCA analysis, I recommend presenting the scree plot.

Line 308-311: It is interesting that young people needed more time to recover stability after perturbations than old people. I did not read any discussion about this finding.

Table 2: For the first row, could the authors add some explanation about these conditions? Or categorize these conditions into anterio-posterior or medio-lateral perturbation. And please add full names of SL and SW.

Table 3: Please make the decimal places consistent.

DISCUSSION

I hope authors could review more relevant studies to discuss the current results. First, although there were no significant differences between 2 groups, or between single and dual tasks, discussion about the non-significance is still needed. I believe there are some studies showing the similar results and providing some potential explanation. Why do the age factor and cognitive loading not significantly influence the recovery time?

Second, the based on the current result, it seems like step length and step width are not sensitive enough to detect the difference between young and old people when they are responding to unexpected perturbations. Could you put more space on justifying the current method and what strength this new way is?

Finally, the part of discussing algorithm advantages, I think the authors could put the description in the supplementary material back in the main text.

Reviewer #2: This study developed a novel algorithm to determine the total recovery time values for regaining stable step length and step width following the different perturbations. This new algorithm might bring some novel to this filed if its reliability could be verified. However, the rationale in this paper is very flawed. As the recovery time value is not a very common variable for gait analysis, it is unclear why such a novel algorithm for assessing recovery time is important. Comparing to the total recovery time proposed by the authors, the initial response time used in many other studies might be more important. In general, the manuscript needs much more work in verifying its statistics, in strengthening its rationale, in modifying its data analysis and in reorganizing the discussion.

Major comments

1. In the introduction, the rationale for estimating the total recovery time of gait parameters is very flawed. Although the authors mentioned that no systematic method for recovery time estimation has been proposed yet, it is still unclear why this total recovery time is crucial for gait analysis or fall prevention. In other words, most studies are directly focusing on the step length/step width/gait speed, why should we take into account this recovery time values for taking a recovery step? What is the advantage of these variables proposed in this study comparing to the others?

2. If this recovery time algorithm is a novel method, the authors should first verify the accuracy of their method by comparing this method with previously proposed methods. At least, they should compare it to the manually detected recovery time. Otherwise, how can we believe the reliability of this method?

3. The main finding in this study was that there were significant differences in the recovery times for either step length or step width between AP and ML perturbations, however, previous studies (i.e. M Zadravec) have already compared the step length, step width, and step time cross perturbations in different directions, and similar conclusion was proposed in these previous studies. What is the novel for this study? Additionally, this new method didn’t show any advantage comparing to the traditional methods (step length/step width/step time), as the same conclusion could be reached using the traditional methods.

4. There were several methodological concerns which questions the results presented and the interpretations the have been highlighted below

Perturbations in 3 different phases were applied, it is unclear why the authors chose these phases, any rationale or reference? Also it is unclear how the relevant gait phases were detected, if you used force plates, please describe how many force plates were used and their position. How was the data used to trigger the perturbations? Wasn’t there a lag time between detection and trigger of perturbation? How was this controlled for? In the method sections, they mentioned that the perturbation occurred after initial contact, mid-stance, or toe-off. What about the results for perturbations after initial contact, and toe-off? For Table 2, it is unclear why the authors only showed the results for perturbations in the mid-stance phase.

It is unclear why the number of gait trials was so different for each participant. Also it seems that every participant was not provided with the same protocol in terms of direction and intensity of perturbation. The authors mention nothing about at which perturbation intensities were the trials collected? The data presented in Table 2 don’t take into account the perturbation intensity. It is known that recovery responses are dependent on perturbation intensity and the authors completely fail to address this. Also the authors mention possible combinations of direction, gait phase and leg to come up to 18 but in my calculation it would be 12 (3x2x2)?

About determining the recovery time threshold. It is not clear what the exact threshold is used in this study to determine a stable pattern or a “recovery”? Neither there is a definition of “stable” nor what a “recovery” actually can be defined as. In the algorithm, firstly a moving window of 6 samples was chosen, and then a moving window of 20 values was used. Would the use of these moving windows lower the sensitivity of the method? How did the author decide the length of the moving window? Please explain.

PCA was used to explore the effects of perturbation direction, group and cognitive load on total recovery times, however, it is unclear what the input matrix is for the PCA algorithm. Please describe this matrix.

Before running the PCA, was the input data rescaled to unit variance?

Did the authors use an independent t-test to compare the extracted PCs between different conditions? If yes, please claim this in the statistics. PCA is a procedure to convert a set of possibly correlated variables into a set of principal components, while it seems there were only two different variables inputted into the PCA in this study. Which means they convert 2 variables into 2 PCs, is it necessary to do so? Why the authors did not directly use an independent t-test to compare the recovery times between different conditions?

Minor comments

1. P3 Line 55, there is an extra bracket.

2. P7 Line 134, for ML perturbation, is the moving platform always moving towards one direction? Please be clear.

3. P8 Line 160, foot marker was mainly used in this study, did the authors do any filtering before using this foot marker? Such as lower-pass filtering?

4. P10 Line 206, for step width calculation, is the trailing foot the same with the recovery foot? Could the recovery foot located anterior to the slipping foot?

5. Figure 2, what’s the duration between the two black dash lines? Why the speed decrease a little in this period?

6. Table 3, what’s the standardized mean? How do you calculate this value? What’s the z value for PC1, and the t value for PC2? Why the authors used different variables for PC1 and PC2 here?

Reviewer #3: Overall, I find the manuscript to be a generally good introduction to a useful methodology that can be used to identify full recovery time from a perturbation during walking. The methodology is novel and generally straight forward and appears to be effective in identifying differences between various experimental conditions.

Abstract

No issues

Introduction

Lines 53-73 – some mention of the methodology used by these authors is missing as they present a fairly comprehensive approach to measuring perturbation recovery during walking - Lee, B.C., Martin, B.J., Thrasher, T.A., Layne, C.S. The Effect of Vibrotactile Cuing on Recovery Strategies From a Treadmill-induced Trip. IEEE Transactions on Neural Systems and Rehabilitation Engineering, DOI 10.1109/TNSRE.2016.2556690.

Lines 77 – 82 – the section on cognitive load and postural control, particularly during walking needs further developed. A quick check of pubmed using ‘cognitive load, walking’ returned 165 articles. Given cognitive load is an independent variable in the study, that section of the introduction needs fleshed out.

Methods

Table 1 – either there is a typographical error regarding BMI or the differences in BMI are so great that comparing across the young and the elderly in balance recovery measures would be inappropriate given the weight differences. Based on the table, I did a rough calculation by converting 169 cm to 67 inches, 64 Kg (young) to 141 lbs and using a BMI chart estimated that the young subjects had an average BMI about 22 which is not far from what is reported in the table. For the elderly, I converted 78 Kg into 172 lbs which resulted in a BMI of approximately 27. This makes me think the BMI of 37.35 reported in the Table is inaccurate. If it is accurate, the average weight of the elderly, given their average height would be over 236 lbs.

Line 134-135 - Two types of perturbations were implemented: (1) medio-lateral platform perturbations were achieved by displacing the moving platform 15 cm over 0.92 seconds.

I read the Figure 2 legend as perturbation times ranged from 1.36 to .6 seconds which leaves me confused as I thought lines 134-35 indicated perturbation time was 0.92 seconds. Please clarify.

Lines 144-145 ‘Four of the twelve older adults were unable to manage this magnitude, thus lower magnitudes were used (levels 3-8, see Fig 2 legend).’ What is meant by unable to manage? Does that mean they were unable to recover from high magnitudes or something else?

Lines 164-165 – ‘Gait cycle phases were detected using foot marker and force plate data.’

A little more detail about how marker and kinetic data were used to obtain gait cycles as there are a variety of ways these measures can be combined to determine gait cycle events.

Lines 176-177 - ‘Four to twelve gait trials were introduced for each participant, each lasting five or ten minutes.’

This statement is confusing as it implies different participants performed different activities. Why would some participants experience four gait trials while others experienced twelve?

Results

No issues

Discussion

There needs to be mention of the Lee et al findings during this discussion. Given their results, to ignore that paper seems a significant oversight. Although different methodology, much of the underlying data (kinetics and kinematics related to tripping) is the same, but they reach somewhat different conclusions regarding recovery time and a brief discussion of why the differences between your data and theirs needs to be included.

Additional discussion of why the authors found differences in M-L recovery versus no differences in A-P recovery should be addressed. Does this relate to the nature of the perturbation, biomechanical and possibly underlying neurophysiological mechanisms, experience, or some interaction of the proceeding?

6. PLOS authors have the option to publish the peer review history of their article (what does this mean?). If published, this will include your full peer review and any attached files.

Reviewer #1: No

Reviewer #2: No

Reviewer #3: No

---

## [Author Response · Author response to Decision Letter 0]

13 Apr 2020

Response to reviewers PONE-D-19-31620

We thank the reviewers for their review of our manuscript and for the constructive comments and suggestions. Herein we address all the suggestions and concerns raised. The manuscript was revised accordingly, a process which ended up with extensive revisions. 

We believe that this process has improved the quality of the manuscript and are grateful for the important feedback. 

Below we summarize each of the reviewers' comments, our response indicating the changes made in the revised manuscript to address the comments. 

Manuscript title: Novel methodology for assessing total recovery time in response to unexpected perturbations while walking

Reviewer 1

The authors of this manuscript developed a new algorithm to estimate the response time after various types of perturbations and compared the differences between young and old adults with principal component analysis (PCA). However, there are still some questions that need to be clarified:

INTRODUCTION 

Comment #1 

 I suggest authors cited some references about using center of mass (COM) and center of pressure (COP) to assess postural stability. Recommended references:

• Fino, Peter C., Ahmad R. Mojdehi, Khaled Adjerid, Mohammad Habibi, Thurmon E. Lockhart, and Shane D. Ross. "Comparing postural stability entropy analyses to differentiate fallers and non-fallers." Annals of biomedical engineering 44, no. 5 (2016): 1636-1645.

• Liau, Ben-Yi, Fu-Lien Wu, Chi-Wen Lung, Xueyan Zhang, Xiaoling Wang, and Yih-Kuen Jan. "Complexity-Based Measures of Postural Sway during Walking at Different Speeds and Durations Using Multiscale Entropy." Entropy 21, no. 11 (2019): 1128. 

Response:

We thank the reviewer for pointing out these sources. In the revised introduction we elaborate on COP and COM referring to concepts used in these and other articles. 

P.3 Line 50-74. Under the section: “Center of pressure and center of mass use for assessing postural stability”

Comment #2 

I suggest the authors mention some limitations about the traditional methods that estimate recovery time in response to unpredictable perturbations. 

Response:

This is now added to the text:” Other algorithms in the literature are specific to one gait parameter (e.g. gait speed [18] or step frequency [20]), which are continuous in their nature and not series of discrete events. 

P.32 line 679-681

Comment #3 

Another concern to me is what the reason the authors compared the differences between the young and old population with PCA. According to the research questions raised in this study, traditional ANOVA or similar statistical methods could be used for this situation. 

Response:

We preferred using PCA from the following reasons: (1) the parameter that we assessed (i.e., total recover time) is not distributed normally; (2) this method is appropriated to analyze multi-dimensional data, looking at the differences in step length and step width at once. With traditional ANOVA we had to compare separately for step length and step width. It is our understanding that the “true” total recovery time is a mix of gait parameters. Further study should evaluate total recovery time (i.e., including the extra-steps that are needed to fully recover gait after balance loss) of a combination of more walking parameters such as cadence, walking speed and so on. 

To address the reviewer comments, we: 

(a) added information about non-normality of the data distribution (legend of Fig. 7); 

(b) added to the text (Data handling and statistical analyses section, lines 435-438) the following: “PCA allows for clustering of multidimensional data, in our case, comparing total recovery time, a bi-dimensional property consisting of the combination of step length and step width, across different groups, conditions and perturbation direction.”.

(c) Non-parametric tests were performed as well. (Wilcoxon rank sum test). These analyses yielded no differences between groups, conditions and perturbation direction (p>0.1). 

Data handling and statistical analyses section, p. 20 lines 435-438

legend of Fig. 7, p. 22 lines 487-489

p.22 lines 466-468

results section p.24-25 lines 524-527

Comment #4 

Line 55: “…a two-stage process [[11] [12]. The…” Please delete the extra left bracket. 

Response:

the whole wording is different now this is not there anymore. 

P. 4 line 83

Comment #5 

Line 59: “first recovery step) [e.g., [13]]. However,…” Please delete “e.g.”. 

Response:

This was deleted, thanks. 

P. 5 line 93

METHODS 

Comment #6 

Line 97: Please add unit for body mass index. 

Response:

this was added in the text. 

p.8 Line 160

Comment #7 

Table 1: Please make the decimal places consistent. 

Response:

This was revised. 

Table 1

Comment #8 

Figure 2: Please write the full name of SSV. 

Response:

This was added in the legend, thank you. 

p.12 line 262

Comment #9 

Line 160: Please specify how many cameras the study used. 

Response:

It is now specified in the text:”18 cameras” 

p.13 line 268

Comment #10 

Line 188: What does the “foot of reference” mean? 

Response:

It is now clarified in the text: “foot of reference (right or left foot that were detected in the relevant gait phase)”. 

p.10 lines 208-209

Comment #11 

Line 234-238: I recommend stating the number of samples of a moving window in text. 

Response:

Was added in the text: ”moving window of 6 samples” 

p.16 line 351

Comment #12 

Line 281-290: Please specify the p value used in this study. 

Response:

This was added:” Significance level was set to p<0.05”. 

p.21 line 458

RESULTS 

Comment #13 

For PCA analysis, I recommend presenting the scree plot. 

Response:

In order not to over load, the figure we added the scree plots in the supplementary material. There is referencing to it in the text (Exploratory analyses: Effects of perturbation type, group, and cognitive load section):” (see scree plots in supplementary material)”.

p.24 lines 519-520

Comment #14 

Line 308-311: It is interesting that young people needed more time to recover stability after perturbations than old people. I did not read any discussion about this finding. 

Response:

Indeed, in the first version it may have been understood that such result was found. In the revised version, in the Discussion, we now emphasize that the differences between age groups were not significant and we discuss it in the light of the existing literature. Further, throughout the discussion we now make it's clearer, that group effect was study only in the framework of a pilot analyses aiming to demonstrate the feasibility of using the total recovery time as defined by the new algorithm. Future work can address group effects on total recovery time. 

p.29-30 lines 626-640

Comment #15 

Table 2: For the first row, could the authors add some explanation about these conditions? Or categorize these conditions into anterio-posterior or medio-lateral perturbation. 

Response:

A row was added on top dividing the perturbations to ML and AP perturbations. The table was moved to the supplementary material. 

Table S1 in supplementary material

Comment #16 

Table 2: please add full names of SL and SW 

Response:

This was added at the legend 

Table S1 in supplementary material

Comment #17 

Table 3: Please make the decimal places consistent. 

Response:

it was revised – Table 3 is now Table 2 

Table 2

DISCUSSION 

Comment #18 

I hope authors could review more relevant studies to discuss the current results. First, although there were no significant differences between 2 groups, or between single and dual tasks, discussion about the non-significance is still needed. I believe there are some studies showing the similar results and providing some potential explanation. Why do the age factor and cognitive loading not significantly influence the recovery time? 

Response:

We reviewed more literature: McIlroy and Maki, 1996; Byrne et al., 2002, Liu and Lockhart et al., 2009, Zwergal et al., 2012, we believe that by doing so the discussion is now enriched, in particular about issues raised by the reviewer. 

We also assert in the revised version that the pilot analyses we conducted regarding age, task, and gait parameters affects, were mainly meant to demonstrate feasibility of the use of the new total recovery time assessment by the objective new algorithm. 

p.29-30 lines 626-648

Comment #19 

Could you put more space on justifying the current method and what strength this new way is? 

Response:

We added relevant paragraphs to the discussion. 

p. 31-32 lines 678-696

Comment #20 

The part of discussing algorithm advantages, I think the authors could put the description in the supplementary material back in the main text. 

Response:

This was moved to the main text. 

p.32 lines 686-696

 

Reviewer 2

This study developed a novel algorithm to determine the total recovery time values for regaining stable step length and step width following the different perturbations. This new algorithm might bring some novel to this filed if its reliability could be verified. However, the rationale in this paper is very flawed. As the recovery time value is not a very common variable for gait analysis, it is unclear why such a novel algorithm for assessing recovery time is important. Comparing to the total recovery time proposed by the authors, the initial response time used in many other studies might be more important. In general, the manuscript needs much more work in verifying its statistics, in strengthening its rationale, in modifying its data analysis and in reorganizing the discussion.

Major comments 

Comment #1 

most studies are directly focusing on the step length/step width/gait speed; why should we take into account this recovery time values for taking a recovery step? What is the advantage of these variables proposed in this study comparing to the others? 

Response:

We thank the reviewer for this question as we herein clarify: The total recovery time values, which are at the focus of the present work, do not refer to a recovery step seen shortly after perturbation is presented. Rather, they reflect the termination of a process of regaining regular gait, i.e., as measured by gait parameters after a perturbation. The importance of our method and findings is the systematic address to that latter process. While it was described that balance is recovered rather quickly after the perturbation (<2 seconds) there are still some recovery processes that are taking place. 

This point is now enhanced in the ‘motivation’ part of the introduction. In addition, we are now mentioning this distinction to the “Algorithm advantages “and “Algorithm engineering advantages” (in the Discussion section) 

p. 4-5 lines 75-100; Walking recovery processes from unexpected perturbations section p. 31-32 lines 678-696in the discussion

Comment #2

If this recovery time algorithm is a novel method, the authors should first verify the accuracy of their method by comparing this method with previously proposed methods. At least, they should compare it to the manually detected recovery time. Otherwise, how can we believe the reliability of this method? 

Response:

ICC analysis was added comparing the algorithm to human labeling. It is now stated in the statistics section:” ICC estimates and their 95% confident intervals were calculated for total recovery time detection using SPSS statistical package version 23 (SPSS Inc, Chicago, IL) based on a mean-rating (k = 2), absolute-agreement, 2-way mixed-effects model, calculated from total recovery time estimation of 100 random perturbations (200 samples – 100 for step length and 100 for step width) comparing the algorithm for total recovery time detections with human labeling (UR, blinded to the results of the algorithm).” The results are presented in the results section:”… moderate reliability (ICC = 0.57, 95% CI = 0.42-0.67, p<0.01 and in the discussion:” In comparison to human labeling the algorithm shows moderate ICC values.” And finally under limitations:” Since there is no ‘gold standard’ for detecting total recovery time we compared the algorithm performance with human labeling, this comparison is limited since human labeling is subjective”. 

Data and statistical analyses 

p.20 lines 426-431

Results p.22 lines 482-483

Discussion p. 26 lines 546-548

p. 33 lines 704-706

Comment #3 

Previous studies (i.e. M Zadravec) have already compared the step length, step width, and step time cross perturbations in different directions, and similar conclusion was proposed in these previous studies. What is the novel for this study? 

Response:

The following references are now included in the revised version: Zadravec’s et al. and other papers we introduce in our introduction section, Lee et al., 2017, Hof et al., 2010 to name a couple, are describing what we refer to as the first recovery process. We see similar behavior in our data to what Zadravec is describing. However, we believe and this is backed by the literature (O’Connor and Donelan [18] and Snaters et al. [20] in our manuscript) that the gait recovery processes continues beyond this point of first recovery of balance. To the best of our knowledge there is no literature about the second process of gait recovery (what we refer to as the total recovery) of step length and step width after physical unexpected perturbations. This is now clarified in the introduction section. 

p. 4-5 lines 75-100

Comment #4 

This new method didn’t show any advantage comparing to the traditional methods (step length/step width/step time), as the same conclusion could be reached using the traditional methods. 

Response:

The advantage of our method is in its ability to cope with any discrete variable such as step width, step length. The methods used to study the second recovery period after the first recovery process (what we refer to as the total recovery time) were tailored specifically to the paradigm in the study in which they were implemented (O’Connor and Donelan [18] and Snaters et al. [20] in our manuscript). To the best of our knowledge step length/step width/step time were all used to describe the first recovery process. It is now emphasized in the introduction and discussion sections. 

Introduction p.5 lines 107-111

Discussion p.31-32 lines 674-696

Comment #5 

Perturbations in 3 different phases were applied, it is unclear why the authors chose these phases, any rationale or reference? 

Response:

We agree with the reviewer the extended introduction of the topic is needed. This is now done in the “Methods” section. Briefly, the rationale behind using all the perturbation combinations was to make them as unpredictable as possible so the participants can’t anticipate the next perturbation to come. Hof and Duysens ([42] in our manuscript) used perturbations in initial contact and toe off, and perturbations in every 10% of the gait cycle. Madehkhaksar et al. ([21] in our manuscript) used anterior-posterior and Medio-lateral perturbations to reduce anticipatory reactions by the participants. We thought initial contact, mid-stance and toe off will provide sufficient variety to negate any learning effects from the perturbation protocol. Citations were added in the manuscript.

p.14 line 302-303

Comment #6 

It is unclear how the relevant gait phases were detected

Response:

Following up on the previous comment we also extended on how gait phases were detected. There was an online procedure based mainly on incoming data from the force plates (see also response to the next comment). We verified that the intended timing of online presentation of perturbation was indeed achieved. This was done by post hoc off line analysis. Both processes are now elaborately described in the Methods section. 

p.11 line 224-238

Comment #7 

if you used force plates, please describe how many force plates were used and their position.

Response:

Description of the force plates installation of properties are now added to the methods section. p.9 lines 183-187

Comment #8 

How was the data used to trigger the perturbations? 

Response:

Please see our response to Comment #6 

p.11 line 224-238

Comment #9 

Wasn’t there a lag time between detection and trigger of perturbation? How was this controlled for? 

Response:

The reviewer raises here an important point. Indeed there was lag of ~250 ms. This issue is now explained in the revised version of the manuscript. Briefly, we re-evaluated the actual perturbation type post-hoc. 

p.11 lines 232-238

Comment #10 

The authors mentioned that the perturbation occurred after initial contact, mid-stance, or toe-off. What about the results for perturbations after initial contact, and toe-off? For Table 2, it is unclear why the authors only showed the results for perturbations in the mid-stance phase. 

Response:

The results for all gait phase are now summarized in two tables (Tables S1 and S2) in the supplementary material.

Table S1 and Table S2 in supplementary material

Comment #11 

It is unclear why the number of gait trials was so different for each participant.

it seems that every participant was not provided with the same protocol in terms of direction and intensity of perturbation. 

Response:

The protocol was consistent across participants, however, due to variability in the participants’ physical capabilities, some participants could not perform the protocol in its entirety and we had to stop before finishing the entire protocol. For example, one young adult participant became noxious and asked to stop. Also, one participant asked to stop due to fatigue (although we instructed our participants that they are allowed to take a rest as much as needed). In the revised version this was clarified in the text. To highlight the planned protocol, we added a figure (Fig.3). 

p.13-14 lines 281-290

Comment #12 

The authors mention nothing about at which perturbation intensities were the trials collected? 

Response:

In the revised methods section under Physical perturbations was now added:” … (perturbation level 12, see Fig. 2)”. 

p.10 line 214

Comment #13 

The data presented in Table 2 don’t take into account the perturbation intensity. It is known that recovery responses are dependent on perturbation intensity and the authors completely fail to address this. 

Response:

In conjunction with our response to the previous comment, only perturbation of the same intensity (i.e., level 12) is now presented in Table S1(Table 2 was moved to the supplementary material and is now Table S1). It is now clarified in the text that there is an effect of perturbation intensity on total recovery time and therefore only perturbation intensity 12 is included in the PCA and the comparisons of total recovery time of group, condition and perturbation direction. Fig.7, histograms were also corrected to contain perturbation intensity 12 only and 95% confidence interval of the median were adjusted as well. 

Table S1 in supplementary material

p.20 lines 438-442

p.22 line 480-481

Comment #14 

 The authors mention possible combinations of direction, gait phase and leg to come up to 18 but in my calculation, it would be 12 (3x2x2)? 

Response:

Herein we explain: “Direction” – 3 – platform to the left or right and treadmill belt deceleration. Gait phase – 3 – Initial contact, mid-stance and toe off. Leg – 2 – right or left. 3*3*2 = 18. This was clarified in the text.

p.10 lines 204-209

Comment #15 

About determining the recovery time threshold. It is not clear what the exact threshold is used in this study to determine a stable pattern or a “recovery”? 

Response:

It’s been emphasized in the method section, Definition of an automated algorithm to detect total recovery time, stage 3 and in the supplementary material: The technique we used is based on the following high level idea: 1. We use the time interval before perturbation to learn what "normal behavior" looks like, 2. We scan the graph from left to right, starting at perturbation time, until we reach a point after which the graph is back to "normal" behavior. 

In the main text we clarified: the algorithm is iteratively evaluating the gains from amplitude differences between an amplitude in the current window, the amplitude in the first window and an amplitude that satisfies a threshold condition (best amplitude; see Eq. 4 for in text). The point of recovery is the point at which the amplitude difference is small enough relative to the ‘distance’ from perturbation. (Eq. 4)

The condition in Eq. 4 is stricter if the position of the current window is further away from the bestamp window index. This is done in order to overcome the natural variance of amplitudes. The point of recovery is defined as the bestind after calculating all amplitude differences in the whole OSDev vector within consecutive windows (see section 4 supplementary material and Fig S1 for complete description of the process of defining the point of recovery). “

It should be clarified that as the algorithm looks for the point in time where the window’s amplitude stops decreasing. This is done by considers how significant is the difference from the previously detected amplitude decrease. For this, we used Eq. 4 and we set a conservative threshold of 1% to prevent undesired delays in total recovery time detection. 

p.18-19 lines 395-412

supplementary material, section 1.4

Comment #16 

There is no definition of “stable” nor what a “recovery” actually can be defined as. 

Response:

We address this point by clarifying the definitions in the text: ”… is considered “recovered”. Meaning having reached a stable pattern - the variability of the gait parameter is reduced to baseline levels. 

p.16 lines 347-349

Comment #17 

In the algorithm, firstly a moving window of 6 samples was chosen, and then a moving window of 20 values was used. Would the use of these moving windows lower the sensitivity of the method? How did the author decide the length of the moving window? Please explain. 

Response:

Our approach to devising the algorithm was by using training and testing sets. The training set was used for the algorithm development including improvement iterations, window sizes, and recovery criterion. The iterations were performed on the training set until the algorithm precision reached ~85% when compared with human labeling. Then the algorithm was applied as is on the testing data and precision was ~78% which is acceptable.

Since the number of data points before the perturbation was limited in the available recordings, we decided to use 25 values before perturbation to define what "normal behavior" means. To define "normal", we used two known "moments" of the graph: (a) local average (b) local standard deviation. Since each of these moments has its own variance even when the graph behaves normally, we collected a distribution of values for each, before the perturbation. To properly represent these distributions (one for avg and one for SD), we thought we need at least 20 values, and since 25 data points were available (see above), we used a window of 6 data points for computing the local average/SD. That way we had 20 moment values: [1,6], [2,7], [3,8].....[20,25].

Note that the window of 6 samples is used to calculate the moments in the original graph while the 20 samples window is used to calculate amplitudes after the perturbation. The amplitude is sensitive to single values, as it computes max(value)-min(value) of the window, so for example, when the exceptionally high value (implied by the perturbation) gets out of the sliding window, a detection might be made, as the new amplitude might significantly decrease. There for we needed a method to define what is significant. This is achieved by applying Eq. 4 in the text. It's true though that the sensitivity of our technique is limited to 6 data points (the inner window). So, on average, our detection might be up to ~3 data points. 

Comment #18 

 it is unclear what the input matrix is for the PCA algorithm. Please describe this matrix.

Response:

Before running the PCA, was the input data rescaled to unit variance? Thank you for this comment: it is now specified in the text: “Prior to using PCA, values of step length and step width were transformed to units of standard deviation so they could be comparable. These values were used in the PCA algorithm as an n x 2 matrix (were n is the length of the standardized step length and step width vectors).”

p.20-21 lines 434-451

Comment #19 

Did the authors use an independent t-test to compare the extracted PCs between different conditions? If yes, please claim this in the statistics. 

Response:

A non-parametric test was used since our data was not normally distribution. Was added:” … using Wilcoxon rank sum test.” 

p. 21 line 454

Comment #20 

The authors convert 2 variables into 2 PCs, is it necessary to do so? Why the authors did not directly use an independent t-test to compare the recovery times between different conditions? 

Response:

Please kindly refer to the reply to comment #3 for reviewer 1 

p. 20 lines 435-438

legend of Fig. 7, p. 22 lines 487-489

p.22 lines 466-468

p.24-25 lines 524-527

Minor comments 

Comment #21 

P3 Line 55, there is an extra bracket. 

Response:

it was Deleted 

P. 4 line 83

Comment #22 

P7 Line 134, for ML perturbation, is the moving platform always moving towards one direction? Please be clear. 

Response:

It is now clarified:”… displacing the moving platform 15 cm, to the left or to the right, over 0.92 seconds.” 

p.10 lines 213-214

Comment #23 

P8 Line 160, foot marker was mainly used in this study, did the authors do any filtering before using this foot marker? Such as lower-pass filtering? 

Response:

Raw data from the foot markers was used – no filters were implemented on marker data. Force plate data was filtered and it is now specified in the text (Methods section). 

p.11 lines 225-226

Comment #24 

P10 Line 206, for step width calculation, is the trailing foot the same with the recovery foot? Could the recovery foot located anterior to the slipping foot? 

Response:

Here in we explain the options:

If a medio-lateral perturbation occurs during initial contact or toe off phase than the trailing limb would be the one to recover the balance and could be placed anterior to perturbed foot. This is the more common reaction; however, some participants chose to place the reacting leg in back of the perturbed foot. In mid-stance perturbation the leading limb (the swinging limb) is the one to recover. For anterior-posterior perturbations: (1) initial contact or toe off phase perturbations the trailing limb will be the recovering foot and it will be placed in back of the perturbed foot. In mid-stance perturbations however, the recovering limb is placed anterior to the perturbed foot.

To clarify this issue in the revised version we modified the caption of Fig.4 and now it reads:

:” Negative step length values represent backward steps of the recovering limb, while negative step width values represent a crossover step with the leading (recovery) limb. 

p.16 lines 337-339

Comment #25 

Figure 2, what’s the duration between the two black dash lines? Why the speed decrease a little in this period? 

Response:

The dashed lines refer to perturbation intensity 12 only. The black lines represent the period of time from the command to initiate perturbation. until the relevant gait phase is detected and the perturbation is executed. In this time the self-paced mode is turned off and the speed is fixed at the average speed of the 5 seconds prior to the perturbation command. The damp in the other graphs is the fixed speed lock by the system. The graphs are synched to the perturbation time. Since the time from the perturbation command to the perturbation execution varies since the system is looking for the right gait phase and the participants has to be in contact with the force plate on the side relevant to the gait phase required as was described in the section about the gait phase detection.

This information is now added to the legend of figure 2. 

Fig. 2 p.12 lines 255-261

Comment #26 

Table 3, what’s the standardized mean? How do you calculate this value? 

Response:

We thank the reviewer for this comment. Table 3 is now Table 2 and Standardized mean is now explained in the text:” Table 2. Mean total recovery time for step length and step width standardized to standard deviation units (standardized mean)”. 

It is calculated separately for step length and step width in the following manner (was added in the text):”Eq.5 ((tRcT-(tRcT) ®))⁄(std(tRcT))

Were tRcT = total recovery time measured in seconds, and std = standard deviation. The standardized total recovery time is unit less.” Table 2 and data handling and statistical analyses.

 p.20-21 line 443 - 451

Comment #27 

Table 3: What’s the z value for PC1, and the t value for PC2? Why the authors used different variables for PC1 and PC2 here? The Z value and t score were the statistics for the Wilcoxson rank sum test and t-test, respectively. 

Response:

The parameters were unified to the non-parametric statistic since PC1 does not have a normal distribution. PC2 distributes normally and that is why we used the 2 different statistics to start with. Table 3 is now Table 2. 

Table 2

Reviewer 3

Overall, I find the manuscript to be a generally good introduction to a useful methodology that can be used to identify full recovery time from a perturbation during walking. The methodology is novel and generally straight forward and appears to be effective in identifying differences between various experimental conditions.

INTRODUCTION 

Comment #1 

Lines 53-73 – some mention of the methodology used by these authors is missing as they present a fairly comprehensive approach to measuring perturbation recovery during walking - Lee, B.C., Martin, B.J., Thrasher, T.A., Layne, C.S. The Effect of Vibrotactile Cuing on Recovery Strategies From a Treadmill-induced Trip. IEEE Transactions on Neural Systems and Rehabilitation Engineering, DOI 10.1109/TNSRE.2016.2556690. 

Response:

Thank you for pointing out this paper: we now introduce Lee et al.’s paper, as well as others. We explain the differences in their approach to ours. Briefly we look at the recovery process after a perturbation as a two stages process. Lee et al. are interested in the first recovery process which they detect using force plate data. We see the same behavior in our data using step width and step length values – these results are not in the scope of this paper.

It is now part of the introduction and the discussion sections. 

Introduction p. 4-5 lines 86-96

Discussion p.26 lines 551-553

Comment #2 

Lines 77 – 82 – the section on cognitive load and postural control, particularly during walking needs further developed. A quick check of pubmed using ‘cognitive load, walking’ returned 165 articles. Given cognitive load is an independent variable in the study that section of the introduction needs fleshed out. 

Response:

We agree with the reviewer on this point. We elaborated on this topic in the introduction section. We choose to concentrate on surveying paradigms addressing the effect of cognitive load on balance recovery from physical perturbations while walking. 

We want to note, however, that the current study does not aim to thoroughly explore the effect of dual tasking on recovery time, but merely to demonstrate feasibility of doing so, based on the small sample size. This is the reason that the effect is not introduced in large details. It is now clarified in the text (discussion section). 

Introduction p.6-7 line 131-140

Discussion p.29 lines 618-625

p.31 lines 672-673

METHODS 

Comment #3 

Table 1 – either there is a typographical error regarding BMI or the differences in BMI are so great that comparing across the young and the elderly in balance recovery measures would be inappropriate given the weight differences. I think the BMI of 37.35 reported in the Table is inaccurate. 

Response:

The calculation was wrong. It was fixed and the p value was adjusted accordingly. 

Table 1

Comment #4 

Line 134-135 - Two types of perturbations were implemented: (1) medio-lateral platform perturbations were achieved by displacing the moving platform 15 cm over 0.92 seconds. I read the Figure 2 legend as perturbation times ranged from 1.36 to .6 seconds which leaves me confused as I thought lines 134-35 indicated perturbation time was 0.92 seconds. Please clarify. The platform was displaced 15 cm over 0.92 seconds. However, some of the older adults asked not to be exposed to this magnitude. and these participants received a lower perturbation magnitude. For the lower magnitudes the displacement was the same (15 cm) but the time period was longer: 1.36 seconds (level 1) to the same displacement over 0.6 seconds (Perturbation magnitude 20). 15 cm over 0.92 seconds corresponds to level 12 – all young adults and most older adults performed at this level.

We now better clarify this point in the revised legend of Fig.2 

p.12 lines 247-252 and Fig. 2 legend

Comment #5 

Lines 144-145 ‘Four of the twelve older adults were unable to manage this magnitude, thus lower magnitudes were used (levels 3-8, see Fig 2 legend).’ What is meant by unable to manage? Does that mean they were unable to recover from high magnitudes or something else? 

Response:

Please see our response to the previous comment (Comment #4). 

p.12 lines 247-252 and Fig. 2 legend

Comment #6 

Lines 164-165 – ‘Gait cycle phases were detected using foot marker and force plate data.’

A little more detail about how marker and kinetic data were used to obtain gait cycles as there are a variety of ways these measures can be combined to determine gait cycle events. 

Response:

Please see our elaborated response to comment #6 Of reviewer 2 on the same topic. 

p.11 line 224-238

Comment #7 

Lines 176-177 - ‘Four to twelve gait trials were introduced for each participant, each lasting five or ten minutes.’

Response:

This statement is confusing as it implies different participants performed different activities. Why would some participants experience four gait trials while others experienced twelve? 

Please see our elaborated response to comment #11 Of reviewer 2 on the same topic. 

p.13-14 lines 281-290

DISCUSSION 

Comment #8 

There needs to be mention of the Lee et al findings during this discussion. Given their results, to ignore that paper seems a significant oversight. They reach somewhat different conclusions regarding recovery time and a brief discussion of why the differences between your data and theirs needs to be included. 

Response:

We agree that it was an oversight on our part. A discussion on the different results was added and explained in our revised discussion. A more elaborate response in comment#1. 

Discussion p.26 lines 551-553

Comment #9 

Additional discussion of why the authors found differences in M-L recovery versus no differences in A-P recovery should be addressed. Does this relate to the nature of the perturbation, biomechanical and possibly underlying neurophysiological mechanisms, experience, or some interaction of the proceeding? 

Response:

Indeed, it is an interesting point. We now discuss/ address aspects mentioned by the reviewer and in the light of general motor control principals regarding the regulation of movement in a broader sense. 

p.30-31 lines 650-671

---

## [Decision Letter · Decision Letter 1]

7 May 2020

Novel methodology for assessing total recovery time in response to unexpected perturbations while walking

PONE-D-19-31620R1

Dear Dr. Plotnik,

We are pleased to inform you that your manuscript has been judged scientifically suitable for publication and will be formally accepted for publication once it complies with all outstanding technical requirements.

With kind regards,

Yih-Kuen Jan, PhD

Academic Editor

PLOS ONE

Additional Editor Comments (optional):

Reviewers' comments:

Reviewer's Responses to Questions

**Comments to the Author**

1. If the authors have adequately addressed your comments raised in a previous round of review and you feel that this manuscript is now acceptable for publication, you may indicate that here to bypass the “Comments to the Author” section, enter your conflict of interest statement in the “Confidential to Editor” section, and submit your "Accept" recommendation.

Reviewer #1: All comments have been addressed

Reviewer #3: All comments have been addressed

2. Is the manuscript technically sound, and do the data support the conclusions?

Reviewer #1: Yes

Reviewer #3: (No Response)

3. Has the statistical analysis been performed appropriately and rigorously? 

Reviewer #1: Yes

Reviewer #3: (No Response)

4. Have the authors made all data underlying the findings in their manuscript fully available?

Reviewer #1: Yes

Reviewer #3: (No Response)

5. Is the manuscript presented in an intelligible fashion and written in standard English?

Reviewer #1: Yes

Reviewer #3: (No Response)

6. Review Comments to the Author

Reviewer #1: (No Response)

Reviewer #3: (No Response)

7. PLOS authors have the option to publish the peer review history of their article (what does this mean?). If published, this will include your full peer review and any attached files.

Reviewer #1: No

Reviewer #3: No

---

## [Editor Report · Acceptance letter]

22 May 2020

PONE-D-19-31620R1 

Novel methodology for assessing total recovery time in response to unexpected perturbations while walking 

Dear Dr. Plotnik:

I am pleased to inform you that your manuscript has been deemed suitable for publication in PLOS ONE. Congratulations! Your manuscript is now with our production department. 

With kind regards,

on behalf of

Dr. Yih-Kuen Jan 

Academic Editor

PLOS ONE